# Purpuramine R, a New Bromotyrosine Isolated from *Pseudoceratina* cf. *verrucosa* Collected in the Kingdom of Tonga

**DOI:** 10.3390/md23050186

**Published:** 2025-04-27

**Authors:** Jennie L. Ramirez-Garcia, Hannah Lee-Harwood, David Ackerley, Michelle Kelly, S. Vailala Matoto, Patricia Hunt, A. Jonathan Singh, Robert A. Keyzers

**Affiliations:** 1School of Chemical & Physical Sciences, and Centre for Biodiscovery, Victoria University of Wellington, P.O. Box 600, Wellington 6140, New Zealand; jenniramirezg@gmail.com (J.L.R.-G.); patricia.hunt@vuw.ac.nz (P.H.); 2Maurice Wilkins Centre for Molecular Biodiscovery, University of Auckland, Private Bag 92019, Auckland 1142, New Zealand; hannahleeharwood@gmail.com (H.L.-H.); david.ackerley@vuw.ac.nz (D.A.); 3School of Biological Sciences, and Centre for Biodiscovery, Victoria University of Wellington, P.O. Box 600, Wellington 6140, New Zealand; 4Oceans Centre, National Institute of Water & Atmospheric Research, P.O. Box 9940, Auckland 1010, New Zealand; michelle.kelly@niwa.co.nz; 5Ministry of Fisheries, Sopu, Vuna Road, Tongatapu P.O. Box 871, Tonga; vailalam@yahoo.com; 6Ferrier Research Institute, and Centre for Biodiscovery, Victoria University of Wellington, P.O. Box 600, Wellington 6140, New Zealand

**Keywords:** bromotyrosine, oxime, computational chemistry, NMR, MS, conformational analysis, antibacterial

## Abstract

Sponges in the verongiid genus *Pseudoceratina* Carter are well-known producers of bioactive secondary metabolites. Chemical screening of a Tongan *P.* cf. *verrucosa* Bergquist using NMR highlighted the presence of aromatic natural products. Subsequent extraction and purification of *P.* cf. *verrucosa* yielded a new bromotyrosine, purpuramine R (**1**), that exhibits moderate (MIC 16 µg/mL) antibacterial activity against Gram-positive *Staphylococcus aureus.* The *E*-geometry of the oxime was confirmed using a combination of NMR and computational approaches. Additionally, computational conformational analysis indicates that purpuramine R adopts a hairpin orientation, stabilized by intramolecular hydrogen and halogen bonds. Knowledge of this stabilized conformation can inform synthetic approaches to make analogues of the purpuramines for future SAR studies.

## 1. Introduction

Marine sponges continue to be a rich resource for the discovery of new bioactive natural products [1]. Within phylum Porifera, the dictyoceratid, dendroceratid, and verongid demosponge orders have yielded numerous marine natural products (MNPs), including ianthelliformisamines [2], bastadins [3], and psammaplins [4]. Whilst the Verongiida are relatively easy to identify in the field due to their rapid oxidation to deep oak brown or deep royal purple upon exposure to air [5,6], the lack of spicules in all three orders makes species identification a challenge. The dense, fleshy, rubbery texture of the Verongiida presents a particular problem; accurate genus and species identification are reliant upon the composition and architecture of the fibers which can be difficult to access without histological examination. More specifically, species of the verongiid genus *Pseudoceratina* are well-known producers of bioactive secondary metabolites. For example, specimens identified in the literature as *P*. *purpurea* (Carter) (which probably includes the more common Indo-Pacific species, *P*. *arabica* (Keller)) are purported to be the source of many new compounds, including merosesquiterpenoids [7,8] and bromotyrosines [9,10].

Exploration of the MNPs from organisms collected in the South Pacific’s Kingdom of Tonga is in its infancy when compared to studies focused on other island groups such as Fiji or Samoa. However, a variety of unique MNPs have been sourced from organisms collected in Tonga [11]. As part of our group’s ongoing examination of Tongan marine organisms for new MNPs [12,13,14,15], we have carried out spectroscopic screening of an extract of *Pseudoceratina* cf. *verrucosa* Bergquist, 1995, a species originally described in New Caledonia [16]. NMR experiments highlighted the presence of aromatic metabolites. Subsequent extraction of a larger amount of *P.* cf. *verrucosa* followed by repeated reversed-phase chromatography resulted in the isolation of a new bromotyrosine metabolite, purpuramine R (**1**, Figure 1). In addition, the known compounds hexadellin A, purealidin B, and purpuramine M were also isolated (Figure 2 and Appendix A) [17,18,19]. The structure of **1** was determined using a combination of NMR and MS techniques, in conjunction with computational approaches to help assign the oxime geometry, and to investigate the conformational space adopted by purpuramine R. Here, we report the structural characterization and bioactivity of purpuramine R (**1**).

## 2. Results

### 2.1. Structure Elucidation of Purpuramine R (***1***)

Purpuramine R (**1**) was isolated as an optically inactive colorless film. Positive-mode high-resolution electrospray mass spectrometric (HRESI-MS) analysis detected a natively charged ion at *m*/*z* 755.8913, suitable for the molecular formula C_24_H_30_^79^Br_4_N_3_O_5_^+^ (calc. *m*/*z* 755.8913, Δ = 0.0 ppm, Appendix A) and therefore requiring ten degrees of unsaturation. The presence of the four bromine atoms in the molecular formula was indicated by the molecular cluster of ions at *m*/*z* 755, 757, 759, 761, and 763 in a 1:4:6:4:1 ratio. Interrogation of the HSQC, HMBC, ^1^H, and ^13^C NMR spectra identified 24 individual carbon resonances, of which 13 were protonated (CD_3_OD; Appendix A). The resonances include aromatic methines, a methoxy group, a series of methylenes, and, of particular note, a singlet ^1^H signal integrating for nine protons, the chemical shifts of which indicated bonding to a heteroatom (δ_C_ 53.6–53.7; δ_H_ 3.20).

Only two spin systems of mutually coupled proton resonances were detected in the COSY spectrum of **1**. The first spin system comprised a pair of methylenes, CH_2_-19 (δ_C_ 28.8, δ_H_ 3.09) and CH_2_-20 (δ_C_ 67.7, δ_H_ 3.54). The chemical shifts of CH_2_-19 and CH_2_-20 indicated attachment to an aryl ring and a nitrogen atom, respectively, supported by appropriate ^1^*J*_C,H_ coupling constants. An HMBC correlation from CH_2_-20 to the deshielded singlet that integrated for nine relative protons indicated the positioning of a terminal trimethyl ammonium moiety. Methylene CH_2_-19 presented HMBC correlations to the aromatic carbon quaternary, C-16 (δ_C_ 136.0), and methine singlet, CH-15/17 (δ_C_ 134.6, δ_H_ 7.58), which integrated for two relative protons, suggesting a symmetrical 1,2,3,5-tetrasubstituted benzene element in **1** (ring B, Figure 3). HMBC correlations from CH-15/17 to a slightly shielded non-protonated aromatic carbon resonance (C-14/18: δ_C_ 119.5) indicated the placement of two of the four bromine atoms of **1**. 

The second spin system of **1** comprised three contiguous methylenes, CH_2_-10 (δ_C_ 37.9, δ_H_ 3.58), via CH_2_-11 (δ_C_ 30.5, δ_H_ 2.10), to CH_2_-12 (δ_C_ 72.2, δ_H_ 4.02), as established from COSY correlations. The chemical shifts of CH_2_-10 and CH_2_-12 suggest the attachment of nitrogen and oxygen, respectively, which is again supported by the ^1^*J*_C,H_ values. Methylene CH_2_-10 correlated to carbonyl C-9 (δ_C_ 166.8), indicating an amide linkage. Additionally, methylene singlet CH_2_-7 also correlated with this carbonyl, along with correlations to a deshielded quaternary carbon, C-8 (δ_C_ 154.9), aromatic quaternary carbons, C-1 (δ_C_ 123.0), and oxygenated C-2 (δ_C_ 151.7), as well as to aromatic methine CH-6 (δ_C_ 134.5, δ_H_ 7.42).

The tyrosine-based moiety of purpuramine R (**1**) was deduced via HMBC correlations from methine H-6 to non-protonated carbons, C-3 (δ_C_ 108.8), C-4 (δ_C_ 154.7). and C-5 (δ_C_ 106.8). The chemical shifts of both C-3 and C-5 were shielded to lower frequencies and established connections to the last two bromines. Carbon C-4 was confirmed to be linked to the methoxy CH_3_-22 (δ_C_ 60.8, δ_H_ 3.80) by HMBC, positioned *para* in relation to C-1 and consistent with the 1,3-dibromo-2-*O*-methoxybenzene (ring A) moieties mostly reported from Verongid sponges. Finally, an HMBC correlation from H_2_-12 to C-13 established the link between the two spin systems via the 1,2,3,5-tetrasubstitutued benzene ring (ring B, Figure 3).

The analysis so far accounted for all but one N, O, and H atom of the molecular formula, indicating that carbon C-8 must be part of an oxime functionality. Such a motif is common in Verongid bromotyrosine metabolites. Assignment of the oxime geometry in bromotyrosines has largely been based upon an empirical observation noted in the 1987 study by Arabshahi and Schmitz, who reported a dimeric bromotyrosine that incorporated both possible geometries. In their example, the carbon chemical shift of the methylene linking the aromatic ring to the oxime was shielded to ~27 ppm for the *E*-isomer, while the *Z*-geometric isomer resonated at ~36 ppm [19]. It should be noted, however, that the substitution pattern around the aromatic rings of their compound were much simpler, being 1,2,4-trisubstituted benzenes rather than the penta-substituted case in **1**. It could be envisaged that electronic shielding effects, particularly from the hydroxy group *ortho* and the methoxy group *para* to the chain attachment, could influence the chemical shift of the methylene group. We therefore sought to use density functional theory calculations of the methylene chemical shifts to help corroborate this empirical approach.

To lower the computational cost, truncated model **2** was used to predict the most stable conformations of the *E* (**2i**) and *Z* (**2ii**) oxime geometries (Figure 4), recognizing that the trimethylammonium portion should not play a significant role in influencing chemical shifts of the tyrosine moiety of **1**. Within each possible geometry, hydrogen bonding between OH-6 and the amide carbonyl, and between the oxime and the amide NH, provides four possible stabilized conformers. Both possible *E*-geometry conformers were found to be lower in energy than the two corresponding *Z*-conformers (Figure 4). Prediction of the NMR chemical shifts of these four conformers (Table 1) showed that the CH_2_-7 methylene chemical shift of purpuramine R matched better to those of the two *E* geometric stereoisomers than the *Z* counterparts; gratifyingly, these also match closely to those observed empirically by Arabshahi and Schmitz [19]. The structure of purpuramine R was therefore established as **1**. The NMR data of **1** are presented in Table 2.

### 2.2. Conformational Analysis of Purpuramine R (***1***)

With the structure of purpuramine R (**1**) in hand, we next took advantage of the computational data already available to spearhead a study of its conformation in solution. We are not aware of any such conformational analyses of bromotyrosine metabolites to have been published to date. Moreover, knowledge of the likely conformation of **1** and related metabolites could inform the rational design of bioactive congeners to investigate structure–activity relationships (SARs) and assist with drug discovery efforts. 

To this end, we employed CREST (Conformer—Rotamer Ensemble Sampling Tool) [20,21] to investigate the conformation space occupied by **1**. Our initial assessment was carried out in the gas phase, which resulted in an ensemble of conformers, the lowest energy of which exhibited a “twisted” structure, stabilized by numerous noncovalent interactions, including hydrogen bonding, π-stacking, and ionic bridging (Figure 5A). To further assess the likely conformation of **1** in solution, CREST calculations were undertaken, employing a generalized solvent model and methanol (MeOH) parameters. The lowest-energy conformer adopts a folded, “hairpin”-like structure (Figure 5B). This solution conformer has a more obvious π-stacking arrangement, generating a pseudo-cyclophane-type arrangement which is “stitched” together by the ammonium moiety interacting with an aromatic bromine. 

It is notable that this folded conformation is very different to the extended planar structure used as standard to depict bromotyrosine MNPs in the literature. It is entirely conceivable that many other related bromotyrosine metabolites will form similar compact folded conformations in solution. Knowledge of such conformational arrangements could also impact the bioactivity of such bromotyrosines. In particular, the more compact folded conformation could significantly impact the mobility of metabolites like purpuramine R through cell membranes or of binding to their receptor targets; therefore, synthetic efforts to make analogues of these metabolites could benefit from a deeper understanding of the three-dimensional structure they possess, enabling the exploitation of unusual functionalities. For example, the formation of a covalent bond between the terminal ammonium unit (to maintain charge carrying capacity) and the tyrosine ring would provide a chemically robust motif that permanently mimics the solution-phase conformation. Alternatively, knowledge of the key noncovalent interactions stabilizing the conformation of **1** could enable the prediction of synthetic modifications that would disrupt intramolecular bonding, providing access to a more flexible extended conformational space, akin to the way researchers normally draw these compounds. Similar conformation analyses of other bromotyrosine metabolites would be informative for future SAR studies of this MNP class.

### 2.3. Bioactivity of Purpuramine R (***1***)

Bromotyrosine metabolites are known to possess antibiotic activity [6,18,22,23,24,25]. Purpuramine R (**1**) was assessed for antibacterial activity against Gram-positive (*Staphylococcus aureus*) and Gram-negative (*Escherichia coli*, *Klebsiella pneumoniae*) strains (Table 3). A moderate effect was observed against *S. aureus* (MIC 16 µg/mL), but no activity was observed against the Gram-negative strains. Testing against a *tolC* deficient strain of *E. coli* (*E. coli* W3110 Δ*tolC*) was also attempted, resulting in moderate inhibition of the bacteria (MIC 16 µg/mL), suggesting that TolC plays an important role in facilitating purpuramine efflux to protect Gram-negative bacteria (Table 3).

## 3. Materials and Methods

### 3.1. General Experimental Procedures

Solvents for purification were obtained from Fisher Scientific (Hampton, NH, USA) and were HPLC or analytical grade. Water used for chromatography was glass distilled prior to use. All solvent mixtures are a reported as % *v*/*v*. Diaion HP-20^®^ was obtained from Supelco (Spruce City, St Louis, MO, USA). Deuterated NMR solvents were obtained from Apollo Scientific (Bredbury, Stockport, United Kingdom). HPLC purifications were performed using an Agilent Technologies 1260 Infinity HPLC system (Santa Clara, CA, USA) featuring a temperature-controlled column compartment, a quaternary pump system, and both evaporative light scattering and diode array detectors. Semi-preparative purifications used a Phenomenex^®^ Luna C18 column (10 mm W × 250 mm L, 5 µm particles; Torrance, CA, USA). UV/vis spectra were extracted from HPLC DAD outputs. High-resolution mass spectra were obtained using an Agilent Technologies 6530 QTOF mass spectrometer coupled to an Agilent Technologies 1260 Infinity HPLC. NMR data (Santa Clara, CA, USA) were acquired using a JEOL JNM-ECZ600R spectrometer (Akishima, Tokyo, Japan) with an N_2_-cooled 5 mm SuperCOOL cryogenic probe (operating at 600 MHz for ^1^H nuclei and 151 MHz for ^13^C nuclei). Chemical shifts (δ, ppm) were referenced to the residual solvent peak [26].

### 3.2. Sponge Material

Sponge specimens were collected from a submerged marine cave on the western coastline of ‘Eua Island in the Kingdom of Tonga (21.40° S, 174.97° W), on 7 June 2016, between 10 to 25 m depth. Specimens were transported frozen to New Zealand and stored at –20 °C until extraction. Identification was performed by author MK who considers the material to be most closely comparable to *Pseudoceratina verrucosa* Bergquist, 1995 (Order Verongiida, family Pseudoceratinidae), a species with a more restricted distribution than the pan Indo-Pacific and closely related species *P. purpurea* (Carter, 1880) and *P*. *arabica* (Keller, 1889). *Pseudoceratina verrucosa* has a more strongly warty surface than the former two species, sand inclusions in the fibers, and prominent oscules. The only other species in genus *Pseudoceratina* is *P. durissima* Carter, 1885, restricted to Australian waters. A Victoria University Wellington (VUW) voucher of the specimen (PTN4_36C) has been accessioned into the NIWA Invertebrate Collection (NIC), NIWA, Wellington: NIWA 143602.

### 3.3. Isolation of Purpuramine R (***1***)

Frozen *P.* cf. *verrucosa* (21.2 g) was macerated in MeOH twice overnight (2 × 50 mL). The second MeOH extract was passed through a HP-20^®^ column (80 mL) that had been pre-equilibrated with Me_2_CO and MeOH (240 mL ea.). The first extract was then passed through the same column and combined with the second extract. The combined extract was diluted with H_2_O (100 mL) and passed back through the same column. Finally, the collected eluent was diluted once more with H_2_O (200 mL; final concentration 25% MeOH_(aq)_) and passed through the HP-20^®^ column. The column was washed with water (240 mL, discarded), followed by 240 mL portions of 30% Me_2_CO_(aq)_, 75% Me_2_CO_(aq)_, and 100% Me_2_CO. The dried 30% Me_2_CO_(aq)_ fraction (35.0 mg) was separated by semi-preparative HPLC (4 mL/min flow rate, 0–3 min: 30% ACN/H_2_O, 0.1% HCOOH; 3–11 min: 60% ACN/H_2_O, 0.1% HCOOH; 11–14 min: 100% ACN, 0.1% HCOOH) to give hexadellin A (t_R_ = 6.8 min, 12.4 mg), purealidin B (t_R_ = 7.7 min, 4.2 mg), purpuramine M (t_R_ = 9.1 min, 4.6 mg), and purpuramine R (1; t_R_ = 10.4 min, 1.4 mg).

Hexadellin A: yellow film; HRESIMS [M + H]^+^ observed *m*/*z* 713.8461, calculated for C_21_H_24_^79^Br_4_N_3_O_5_^+^ *m*/*z* 713.8444; ^1^H NMR (600 MHz, CD_3_OD) δ 7.55 (s, 2H), 6.42 (d, *J* = 0.9 Hz, 2H), 4.10–4.05 (m, 6H), 3.77 (s, 1H), 3.73 (s, 6H), 3.59 (t, *J* = 7.0 Hz, 4H), 3.16 (t, *J* = 7.5 Hz, 4H), 3.09 (s, 1H), 2.90 (t, *J* = 7.5 Hz, 4H), 2.12 (m, 2H); ^13^C NMR (151 MHz, CD_3_OD) δ 161.6, 155.3, 153.6, 149.3, 134.4, 132.2, 122.8, 119.5, 114.2, 92.4, 75.5, 72.2, 60.4, 41.4, 40.1, 37.9, 33.1, 30.6. Data concordant with those in the literature [17].

Purealidin B: colorless film; HRESIMS [M]^+^ observed *m*/*z* 755.8920, calculated for C_24_H_30_^79^Br_4_N_3_O_5_^+^ *m*/*z* 755.8913; ^1^H NMR (600 MHz, CD_3_OD) δ 7.62 (s, 3H), 6.42 (d, *J* = 0.9 Hz, 2H), 4.10–4.04 (m, 5H), 3.77 (s, 1H), 3.73 (s, 5H), 3.62–3.52 (m, 6H), 3.20 (s, 13H), 3.13–3.07 (m, 5H), 2.12 (m, 3H); ^13^C NMR (151 MHz, CD_3_OD) δ 161.6, 155.3, 153.7, 149.3, 136.1, 134.6, 132.1, 122.8, 119.4, 114.2, 92.4, 75.5, 72.2, 67.6, 60.3, 53.6, 40.1, 37.9, 30.6, 28.8. Data concordant with those in the literature [27]. 

Purpuramine M: white solid; HRESIMS [M + H]^+^ observed *m*/*z* 713.8431, calculated for C_21_H_24_^79^Br_4_N_3_O_5_^+^ *m*/*z* 713.8444; ^1^H NMR (600 MHz, CD_3_OD) δ 7.52 (s, 3H), 7.43 (s, 1H), 4.03 (t, *J* = 6.0 Hz, 3H), 3.81 (d, *J* = 6.0 Hz, 7H), 3.59 (t, *J* = 6.8 Hz, 3H), 3.15 (t, *J* = 7.6 Hz, 3H), 2.89 (t, *J* = 7.6 Hz, 3H), 2.10 (quin, *J* = 6.5 Hz, 3H); ^13^C NMR (151 MHz, CD_3_OD) δ 166.6, 154.7, 154.2, 153.3, 151.3, 136.9, 134.3, 134.1, 122.6, 119.3, 108.7, 107.1, 71.9, 60.4, 41.1, 37.7, 32.9, 30.2, 25.1. Data concordant with those in the literature [18]. 

Purpuramine R (1): white solid; UV/vis: λ_max_ 207, 230, 290 nm (spectrum extracted from HPLC DAD); HRESIMS [M]^+^ observed *m*/*z* 755.8913, calculated for C_24_H_30_^79^Br_4_N_3_O_5_^+^ *m*/*z* 755.8913; HRESIMS/MS (30 eV) m/z (% relative intensity) 503.1067 (13), 429.0888 (21), 415.0365 (26), 355.0699 (29), 341.0179 (56), 337.9567 (29), 281.0510 (100), 221.0843 (37), 207.0321 (20), 147.0656 (42); for NMR data, see Table 2.

### 3.4. Computational Chemistry

Density functional theory (DFT) calculations were performed with the G16 suite of codes and visualized with GView6 [28,29]. For all calculations, (a) the density integration grid was set to a pruned (99,590) grid, i.e., 99 radial shells and 590 angular points per shell (int = ultrafine); (b) the scf convergence was set to 10^−9^ (scf = conver = 9) for optimization and 10^−10^ for frequency and NMR calculations. A value of 10^−x^ corresponds to a maximum of 10^−x^ on the RMS density matrix, with 10^−(x−2)^ on the maximum value of the density matrix and the energy. Optimized conformers have been confirmed as the minima by the absence of imaginary modes.

A scaling method described on CHESHIRE CCAT (the Chemical Shift Repository for computed NMR scaling factors, with Coupling Constants Added Too) was employed [30,31]. Initial optimizations were carried out in the gas phase (GP) at the B3LYP(D3BJ)/6-311 + G(d,p) level. The lowest-energy conformers (A-D) were subsequently optimized in the GP at the B3LYP(D3BJ)/6-31 + G(d,p) level employing a Lanl2DZdp pseudo-potential (PP) and associated basis-set for the Br atoms. Single-point NMR evaluations were then carried out at the mPW1PW91-PCM(methanol)/6-311 + G(2d,p) level. Log files are available at DOI: 10.5281/zenodo.15098754

The NMR chemical shifts were scaled based on data given in CHESHIRE CCAT, as shown in eqn 1 (where σ_iso_ =the isotropic SCF GIAO Magnetic shielding tensor in ppm), and the RMSD given is for a test set of molecules [24,30]. We note that the D3BJ correction has been employed at the B3LYP level relative to the geometry optimization method defined in CHESHIRE CCAT B3LYP/6-31 + G(d,p). This change in method means that the fitting Equation (1) may not be as accurate as is indicated by the RMSD reported in CHESHIRE CCAT (2.0632). However, an error of ≈2 ppm in ^13^C NMR chemical shifts is expected.^13^C s = −1.0399 σ_iso_ + 186.5993 RMSD = 2.0632(1)

The initial B3LYP-optimized structures were first re-optimized using the XTB program (Semiempirical Extended Tight-Binding Program Package) employing the default options and using the Geometry, Frequency, Noncovalent, eXtended TB (GFN2-xTB) algorithm [32,33]. The CREST software (Conformer–Rotamer Ensemble Sampling Tool, v2.12) generates conformer/rotamer ensembles though meta-dynamic (MTD) simulations and sampling and employs a genetic z-matrix crossing algorithm (GC) [20,21]. CREST was run using the GFN2-xTB algorithm, first in the gas phase, and then including a generalized solvent description for methanol (ALPB). The iMTD-GC workflow was employed using default options: RMSD threshold 0.125, population threshold 0.05, conformer energy window 6 kcal/mol.

A “linear” bromotyrosine diagram (shown as **1**) was constructed and optimized in the GP at the HF/3-21G level. The HF/3-21G structure was then optimized in the GP at the XTB level and a CREST conformer calculation carried out, 991 conformers/rotamers were obtained. The lowest-energy conformer was used to start a second CREST search which generated 1684 conformers/rotamers. The lowest-energy structure of the second search is very similar to that obtained in the first search. The HF/3-21G GP structure was also optimized within a generalized PCM solvent environment for MeOH at the XTB level and a CREST conformer calculation carried out, and 1470 conformers/rotamers were obtained and the lowest-energy conformer was retained for analysis. The lowest-energy XTB conformers for the GP and MeOH are given in Figure 5. 

### 3.5. Bioassays

MIC (minimum inhibitory concentration, the lowest concentration at which visible growth is completely inhibited) assays were performed using a methodology adapted from Wiegand et al. [34]. Strains were freshly streaked onto Mueller Hinton (MH) agar plates and incubated for 16 h at 37 °C. Following colony formation, 5–10 colonies were resuspended in MH medium, and the OD_600_ of each culture was normalized to 0.002. Cells were seeded in wells containing MHB media supplemented with a final concentration of 3.2% DMSO or drug between 0.125 μg/mL to 32 μg/mL (in 2-fold dilutions). Plates were incubated at 37 °C for 16 h and then visually inspected for visible growth. Assays were performed in biological triplicates.

## 4. Conclusions

Verongid sponges such as *Pseudoceratina* cf. *verrucosa* remain an important resource for the discovery of new bioactive metabolites. This study, guided by an untargeted NMR-based spectroscopic screen of Tongan marine invertebrates, has resulted in the isolation of a new bromotyrosine, purpuramine R (**1**). Structurally, **1** combines the trimethylammonium group found in chiral purealidin B with the free oxime as present in purpuramine M. Many bromotyrosine metabolites have been reported from Verongiid sponges, which have been recently reviewed [35]. The interested reader is directed to this reference to compare the structural relationship between our isolated compounds and other members of the class. Purpuramine R exhibits moderate antibacterial activity against a model Gram-positive strain and was identified as a substrate for TolC-mediated efflux in Gram-negative bacteria. The use of computational chemistry has facilitated verification of the empirical approach commonly employed to establish the geometry of the oxime functional group found in many of these metabolites. Moreover, analysis using CREST has discovered an unexpected three-dimensional conformation stabilized by numerous intra-molecular interactions. We believe that similar computational studies of related metabolites will be informative for informed rational design of the analogs, SAR probes, and future drug leads. 

## Figures and Tables

**Figure 1 marinedrugs-23-00186-f001:**
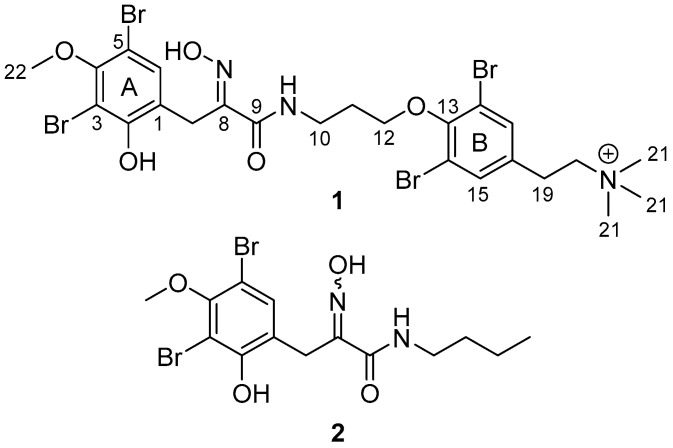
Purpuramine R (**1**) and truncated structure (**2**) used for calculations.

**Figure 2 marinedrugs-23-00186-f002:**
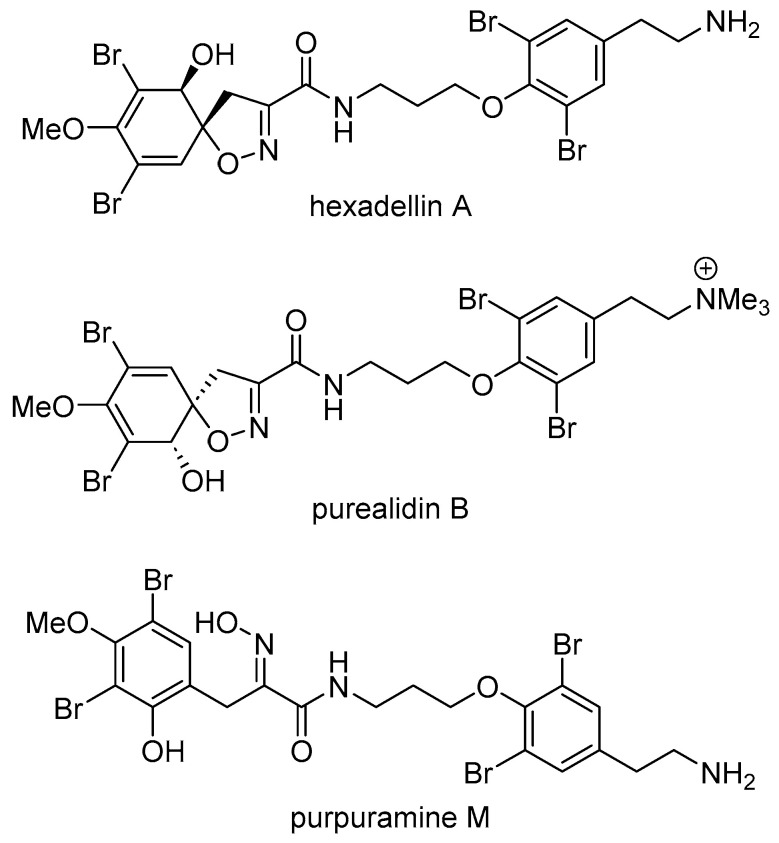
Known compounds isolated from *P.* cf. *verrucosa*.

**Figure 3 marinedrugs-23-00186-f003:**
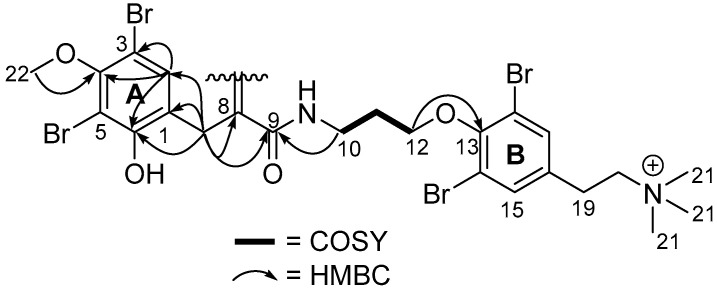
Establishment of main structure of purpuramine R (**1**).

**Figure 4 marinedrugs-23-00186-f004:**
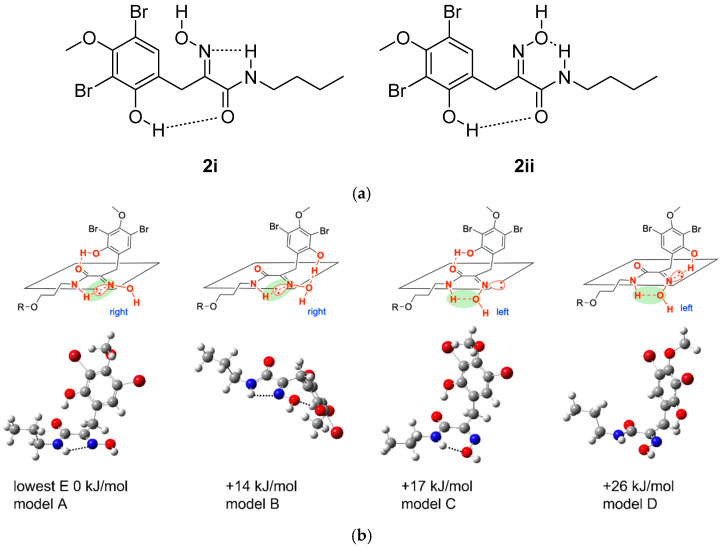
Truncated model **2** for prediction of NMR chemical shifts. (**a**) Two possible hydrogen-bond-stabilized geometries of **2**. (**b**) Predicted energies of hydrogen-bonded conformers of **2i** and **2ii**.

**Figure 5 marinedrugs-23-00186-f005:**
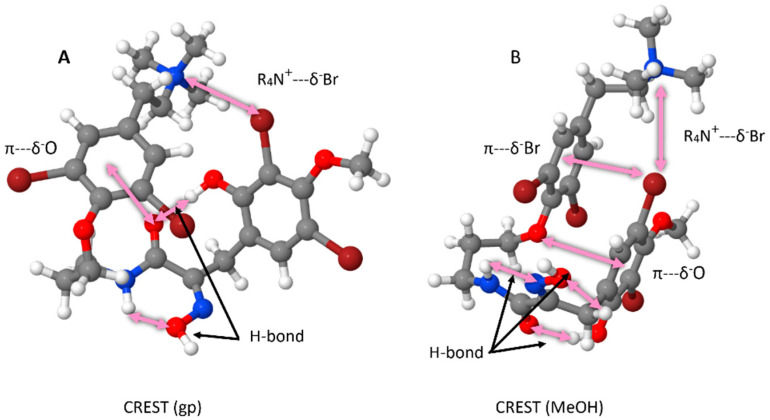
Calculated lowest-energy conformers at the XTB level of purpuramine R (**1**) in the gas phase (**A**) and MeOH (**B**) models.

**Table 1 marinedrugs-23-00186-t001:** Calculated chemical shifts (C-7 to C-10) of **2**.

	Oxime Configuration	C-7	C-8	C-9	C-10	MAE ^a^
Experimental		25.6	154.9	166.8	37.9	0.00
Model A	*E*	25.8	153.6	164.0	42.7	2.33
Model B	*E*	25.5	157.8	160.3	42.3	4.13
Model C	*Z*	34.4	153.0	159.8	44.2	5.89
Model D	*Z*	34.3	154.4	157.3	42.7	6.36

^a^ MAE: mean absolute error.

**Table 2 marinedrugs-23-00186-t002:** ^13^C (151 MHz) and ^1^H (600 MHz) NMR spectroscopic data (CD_3_OD) of purpuramine R (**1**).

Position	^13^C			^1^H		
	δ (ppm)	Type	^1^*J*_C,H_ (Hz)	δ (ppm)	Mult.	*J* (Hz)
1	123.0	C	-	-		
2	151.7 ^a^	C	-	-		
3	108.8 ^a^	C	-	-		
4	154.7 ^a^	C	-	-		
5	106.8 ^a^	C	-	-		
6	134.5	CH	162	7.42	s	
7	25.6	CH_2_	137	3.81	s, 2H	
8	154.9 ^a^	C	-	-		
9	166.8 ^a^	C	-	-		
10	37.9	CH_2_	134	3.58	t, 2H	6.7
11	30.5	CH_2_	126	2.10	quin, 2H	6.4
12	72.2	CH_2_	148	4.02	t, 2H	5.9
13	153.5 ^a^	C	-	-		
14/18	119.5	C	-	-		
15/17	134.6	CH	170	7.58	s, 2H	
16	136.0 ^a^	C	-	-		
19	28.8	CH_2_	128	3.09	m, 2H	
20	67.7	CH_2_	148	3.54	m, 2H	
21	53.6–53.7	N^+^(CH_3_)_3_	148	3.20	s, 9H	
22	60.8	CH_3_	145	3.80	s, 3H	

^a^ Detected by HMBC.

**Table 3 marinedrugs-23-00186-t003:** Antibacterial activity (MIC) of purpuramine R (**1**); µg/mL (µM).

	*S. aureus* ATCC 25923 (µg/mL)	*E. coli* W3110 (µg/mL)	*E. coli* W3110 Δ*tolC* (µg/mL)	*K. pneumoniae* KPLN49 (µg/mL)
Purpuramine R (**1**)	16 (21.1)	>32 (42.2)	16 (21.1)	>32 (42.2)
Gentamicin (positive control)	0.25 (0.52)	0.5 (1.05)	0.5 (1.05)	>32 (42.2)

## Data Availability

Primary NMR datafiles (FIDs) are publicly available from the Harvard Dataverse via https://dataverse.harvard.edu/dataset.xhtml?persistentId=doi:10.7910/DVN/LTJFAX, accessed 14 January 2024. The geometry XYZ and CREST computational data are available via DOI: 10.5281/zenodo.15098754.

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
