# Peer review of "Purpuramine R, a New Bromotyrosine Isolated from Pseudoceratina cf. verrucosa Collected in the Kingdom of Tonga"

_marinedrugs, 2025, doi:10.3390/md23050186_

Round 1
Reviewer 1 Report
Comments and Suggestions for Authors
The manuscript is very well written. I have only one suggestion for improving it before publication - it is necessary to discuss the novelty of purpuramine R, to indicate how it differs from other related compounds.
Author Response
The manuscript is very well written. I have only one suggestion for improving it before publication - it is necessary to discuss the novelty of purpuramine R, to indicate how it differs from other related compounds.
Response: We are sincerely grateful for this reviewers kind words. We have have added a short section and reference to the conclusions section that we hope addresses this request adequately. The text is:
Structurally, 1 combines the trimethylammonium group found in chiral purealidin B with the free oxime as present in purpuramine M. Many bromotyrosine metabolites have been reported from Verongiid sponges, which have been recently reviewed [35].
Reviewer 2 Report
Comments and Suggestions for Authors
The manuscript entitled “Purpuramine R, a New Bromotyrosine Isolated from Pseudoceratina cf. verrucosa Collected in the Kingdom of Tonga” describes the chemical screening of a Tongan Pseudoceratina cf. verrucosa (Bergquist) using NMR techniques, which led to the discovery of a new bromotyrosine, purpuramine R (1). This compound exhibits moderate antibacterial activity with MIC value of 16 µg/mL against Gram-positive Staphylococcus aureus. The authors also applied NMR and computational methods to determine the E-geometry of the oxime moiety. They propose that purpuramine R adopts a hairpin conformation, stabilized by intramolecular hydrogen and halogen bonding, as supported by their computational conformational analysis. They suggest that understanding this stabilized conformation could inform synthetic efforts to develop purpuramine analogues for future structure-activity relationship (SAR) studies.
The manuscript is well-structured and clearly written. However, the discovery of purpurealidin-type bromotyrosines has been extensively explored in the late 1990s and early 2000s. These compounds are already known for their cytotoxic and antimicrobial activities. While the authors argue that their computational findings could support future synthetic efforts, similar work has already been published; for example, the 2018 Marine Drugs paper (Mar. Drugs 2018, 16(12), 481; https://doi.org/10.3390/md16120481), which reported the chemical synthesis of purpurealidin I and its derivatives, including 33 purpurealidin-inspired simplified amides, and their SAR toward antiproliferative activity.
Therefore, despite the nice experimental and computational work, the novelty and biological significance of this discovery are limited. There is little new chemistry or biology presented that would meet the publication threshold for Marine Drugs. I recommend the authors consider submitting this manuscript to a different MDPI journal, such as Molecules, which might be a more appropriate journal for this work.
Author Response
"...despite the nice experimental and computational work, the novelty and biological significance of this discovery are limited. There is little new chemistry or biology presented that would meet the publication threshold for Marine Drugs. I recommend the authors consider submitting this manuscript to a different MDPI journal, such as Molecules, which might be a more appropriate journal for this work."
Response: We thank the reviewer for the generally positive view of the content of our manuscript, even though ultimately they recommend publication elsewhere. We can understand the position that they have taken but are grateful that the other reviewers and the editor have deemed our manuscript worthy of publication in Marine Drugs. Nevertheless we thank the reviewer for the time that they have taken to provide their summary.
Reviewer 3 Report
Comments and Suggestions for Authors
The paper reports a new bromotyrosine from P. cf. verrcosa. The structure was elucidated using NMR and computational approaches. The antibacterial activity of the new isolate was also evaluated, which showed moderate activity. Overall, the structure is interesting and the manuscript is well-written. Minor comments:
- Did the author run an LC-MS of the whole extract? Are there any other kinds of compounds?
- For a better reading, the author should also show the structures of the three known compounds along with the new compound.
- There are missing lots of carbon signals of compound 1, do you have an explanation or did you try a different NMR solvent?
- In the isolation section 3.3, what is the flow? Please make it in detail.
- Figure 9 is unclear, what do the colored lines mean?
Author Response
The paper reports a new bromotyrosine from P. cf. verrcosa. The structure was elucidated using NMR and computational approaches. The antibacterial activity of the new isolate was also evaluated, which showed moderate activity. Overall, the structure is interesting and the manuscript is well-written.
Response: We are sincerely grateful for the reviewer's efforts in examining our manuscript, and for their positive view of our work.
Minor comments:
1. Did the author run an LC-MS of the whole extract? Are there any other kinds of compounds?
Response: Actually, no, we did not. This project was part of a NMR spectroscopy-guided study that sought to use alternative NMR experiments for screening, and we wished to focus attention on that application only. We purposefully avoided any LCMS analyses other than obtaining suitable HRMS data for structure elucidation. We did not note any NMR-based spectroscopic evidence of non-bromotyrosine metabolites. However, we cannot rule out the presence of other minor metabolites in the extracts.
2. For a better reading, the author should also show the structures of the three known compounds along with the new compound.
Response: We agree with this comment and have added a new figure 2 to provide that information. We have renumbered all other figures accordingly.
3. There are missing lots of carbon signals of compound 1, do you have an explanation or did you try a different NMR solvent?
Response: With limited sample, we were unable to obtain an unambiguous, direct detected 13C NMR spectrum. Rather, as noted as an existing footnote to table 2, some of the 13C NMR chemical shifts were obtained from indirect detection via HMBC.
4. In the isolation section 3.3, what is the flow? Please make it in detail.
Response: The flow rate for the HPLC purification is listed in the general experimental conditions part, section 3.1, for the submitted manuscript. We have moved that particular information to section 3.3 as requested.
5. Figure 9 is unclear, what do the colored lines mean?
Response: We appreciate the referee's concerns. Figure S9 is a trace from our HPLC system for compound 1. The different lines represent the various normalization data channels and resulting UV/vis traces. Unfortunately, a data breach means we no longer have the raw data files to reprocess and only obtain the one overall UV/vis spectrum. Considering this, we are happy to remove figure S9 from the SI file if the Editor deems this image too confusing.
Reviewer 4 Report
Comments and Suggestions for Authors
The authors presented the structure of purpuramine R, a new bromotyrosine isolated from Pseudoceratina cf. verrucosa. The content of this manuscript is well organized. This manuscript contains content that is of interest to experts in this field as well as non-experts. To make this manuscript even better, please consider the following comments.
- Please include the structures of the known compounds purpuramine M, purealidin B, and hexadellin A in the manuscript, as this will be of benefit to readers of this journal.
- Line 139; The authors have used computational chemistry techniques to predict the conformation of the new compound 1. The authors should verify the predicted conformation of 1 from data from NOE or ROE experiments.
- Table 2 and supplementary materials; In the 1H-NMR spectrum of compound 1 included in the supplementary materials, a broad singlet signal is observed between 8.4 and 8.6 ppm. However, the authors do not mention this signal in Table 2. I speculate that this signal is a hydrogen signal of "NH". And, I think that this signal is observed in the low magnetic field through a hydrogen bond such as 2i in Figure 3. Furthermore, this broad signal is also observed in the 1H-NMR spectrum of the known compounds hexadellin A, purealidin B, and purpuramine M included in the supplementary materials. This signal may support the geometric isomerism of oxime. The authors should mention the above point.
Author Response
The authors presented the structure of purpuramine R, a new bromotyrosine isolated from Pseudoceratina cf. verrucosa. The content of this manuscript is well organized. This manuscript contains content that is of interest to experts in this field as well as non-experts. To make this manuscript even better, please consider the following comments.
Response: Again, we sincerely appreciate the time and effort this reviewer has taken to give such a positive response to our paper.
1. Please include the structures of the known compounds purpuramine M, purealidin B, and hexadellin A in the manuscript, as this will be of benefit to readers of this journal.
Response: As noted for a previous reviewer, we have added additional figure 2 to address this concern.
2. Line 139; The authors have used computational chemistry techniques to predict the conformation of the new compound 1. The authors should verify the predicted conformation of 1 from data from NOE or ROE experiments.
Response: We thank the reviewer for this technically sound request. Unfortunately, we did not expect the hairpin conformation that ultimately was suggested by the computational experiments. In the intervening time, while the calculations were being made, we used up all the compound isolated for the bioassays. Given the compound is non-chiral, we had neglected to obtain ROESY/NOESY data. We are therefore unable to comply with this request, even though we agree that it would be excellent experimental evidence to support the computational conformational proposal.
3. Table 2 and supplementary materials; In the 1H-NMR spectrum of compound 1 included in the supplementary materials, a broad singlet signal is observed between 8.4 and 8.6 ppm. However, the authors do not mention this signal in Table 2. I speculate that this signal is a hydrogen signal of "NH". And, I think that this signal is observed in the low magnetic field through a hydrogen bond such as 2i in Figure 3. Furthermore, this broad signal is also observed in the 1H-NMR spectrum of the known compounds hexadellin A, purealidin B, and purpuramine M included in the supplementary materials. This signal may support the geometric isomerism of oxime. The authors should mention the above point.
Response: Unfortunately this resonance is from residual formic acid used in the HPLC separation of the compounds, hence its consistent appearance in all 1H NMR spectra.